# Starch Physicochemical Properties of Normal Maize under Different Fertilization Modes

**DOI:** 10.3390/polym15010083

**Published:** 2022-12-25

**Authors:** Jue Wang, Dalei Lu

**Affiliations:** Jiangsu Key Laboratory of Crop Genetics and Physiology, Jiangsu Key Laboratory of Crop Cultivation and Physiology, Jiangsu Co-Innovation Center for Modern Production Technology of Grain Crops, Yangzhou University, Yangzhou 225009, China

**Keywords:** slow-release fertilizer, starch structure, pasting viscosity, retrogradation

## Abstract

Improving the quality with desired functions of natural starch through agronomic practice will meet the increasing need of people for natural, functional foods. A one-off application of slow-release fertilizer is a simple and efficient practice in maize production, though its influence on the starch quality is scarce. In the present study, the structural and functional properties of the starch of normal maize under two fertilization modes (one-off application of slow-release fertilizer at the sowing time (SF), and three applications of conventional fertilizer at the sowing time, and topdressing at the jointing and flowering stages (CF)) under the same fertilization level (N/P_2_O_5_/K_2_O = 405/135/135 kg/ha) were studied using Jiangyu877 (JY877) and Suyu30 (SY30) as materials. The observed results indicate that the size of starch granules was enlarged by fertilization and the size was the largest under CF in both hybrids. The amylose content was unaffected by CF and reduced by SF in both hybrids. In comparison to no fertilizer (0F), the peak 1/peak 2 ratio was decreased by CF in both hybrids, whereas the ratio under SF was unaffected in JY877 and decreased in SY30. The amylopectin average chain-length was reduced by fertilization and the reduction was higher under CF in JY877. The relative crystallinity was increased by CF in both hybrids and the value under SF was unaffected in SY30 and increased in JY877. The peak, trough, and final viscosities of starch were increased by fertilization in both hybrids. The starch thermal characteristics in response to fertilization modes were dependent on hybrids. The retrogradation enthalpy and percentage were increased by CF in both hybrids, whereas those two parameters under SF were increased in SY30 and decreased in JY877. In conclusion, starch with similar granule size, higher peak 1/peak 2 ratio, and lower relative crystallinity was obtained under SF than under CF for both hybrids. Longer amylopectin chain-length was observed in JY877, which induced lower pasting viscosities in SY30 and lower retrogradation characteristics in JY877.

## 1. Introduction

Among different starch resources, maize is the most important and dominates about 80% of the global market share [1]. In the traditional food industry, the starch quality is often modified by some enzymatic or chemical methods. With improving living standards, natural starch of a desired quality has received extensive attention. The natural starch quality affects its utilization and is dependent on genotypes, environmental factors, and agronomic practices. Among various agronomic practices, nitrogen (N) fertilization is an important measure and N type, application rate, time, and mode affect the starch quality. In maize production, N fertilization induces a larger effect for protein content as N rates increase, whereas the contents of starch and oil present less varied N rates [2]. Higher N rate in four to five split applications increases maize grain quality and yield [3], and a higher N rate increases oil, total unsaturated fatty acids, and starch and amylopectin under high irrigation level, though it does not affect grain quality under low irrigation level [4]. With an increase of the N rates, the contents of resistant starch, amylose, and phytic acid in maize also increase [1]. The waxy maize starch pasting and thermal properties are affected by N fertilization and starch with large starch granule size (SGS) exhibit high trough (TV), final (FV), and setback (SB) viscosities and gelatinization temperatures [5]. Studies on wheat [6,7], common buckwheat [8,9], Tartary buckwheat [10,11], and rice [12,13,14,15,16,17,18] have reported that starch pasting and thermal qualities of cereal grain are affected by N rates, split times, application stages, and the ratio of base fertilizer to topdressing application via changes in the grain composition contents and starch structures such as SGS, amylopectin chain-length (ACL), and relative crystallinity (RC). 

With the development of the science and technology of agronomy, the use of fertilization transfers to simultaneously improve grain yield and quality, and the use of slow-release fertilizer (SF) as an efficient and environment friendly fertilizer have been widely adopted in maize and other cereal crops production. The N release of the SF agrees with the crop nutrient requirement, allowing for the coordination of vegetative and reproductive growth, a reduction in nutrient loss, an improvement in N utilization efficiency and a reduction in topdressing labor costs [19]. In the past decade, studies on rice [20,21,22,23,24] and wheat [25,26,27,28,29] have reported that the application of SF affects the grain quality by changing the concentrations of protein, starch, oil and mineral elements. However, studies have mainly focused on the grain quality while starch structural and functional properties have received less attention [30,31]. 

A one-time application of SF is a low labor-cost practice in modern agriculture, and grain quality has received more attention as living standards have improved. However, questions regarding the effects of split applications of traditional compound fertilizers and urea and one-off application of SF on the quality of normal maize starch remain unanswered. In this study, the physicochemical properties of starch under two fertilization modes (one-off application of slow-release fertilizer at the sowing time, and three applications of conventional fertilizer at the sowing time, topdressing urea at both the jointing and flowering stages) of two normal maize hybrids are compared. 

## 2. Material and Method

### 2.1. Experimental Design

The experiment was conducted in Jiangxinsha farm (N 31°48′, E 121°05′), Nantong, Jiangsu, China, in 2018. The soil (0–20 cm) nutrient content before sowing was as follows: 11.83 g/kg organic matter, 1.11 g/kg total N, 81.36 mg/kg available N, 6.31 mg/kg available P, and 61.49 mg/kg available K. Pests, diseases, and weeds were controlled by managers. Cumulative temperature, rainfall, and sunlight during plant growth were 2925.8 °C, 290.2 mm, and 670.6 h, respectively. 

Two normal maize hybrids widely planted in Jiangsu, namely Jiangyu877 (JY877) and Suyu30 (SY30), were used as the materials. Seeds were sown on 1 April with a density of 75,000 plants/ha, and both hybrids were harvested on 6 August. A field trial was performed with a randomized complete block design with three replications. All plots were 45.0 m long and 3.6 m wide with row spacing of 0.4 and 0.8 m.

The fertilization modes included no fertilizer (0F), one-off application of slow-release fertilizer at the sowing time (SF), and three applications of conventional fertilizer at the sowing time, jointing stage and flowering stage (CF). The fertilization rates of N, P_2_O_5_ and K_2_O were 405, 135 and 135 kg/ha, respectively, for both CF and SF [32]. SF (N/P_2_O_5_/K_2_O = 27%/9%/9%) (Lvjuneng, Zhongdong, Changzhou, China) was applied at the rate of 405 kg/ha N (P_2_O_5_ and K_2_O were 135 kg/ha) at the sowing time as basal fertilizer. CF was applied at the rate of 135 kg/ha N, with P_2_O_5_ and K_2_O forming a conventional compound fertilizer (N/P_2_O_5_/K_2_O = 15%/15%/15%) that was used as a basal fertilizer at the sowing time, an N rate of 225 and 45 kg/ha was used for conventional urea (N = 46%) at the jointing and flowering stages as topdressing fertilizer. 

### 2.2. Starch Isolation

Grains (100 g) were steeped in 500 mL of distilled water containing 1 g/L NaHSO_3_ at room temperature for 48 h. Starch was isolated in accordance with a previously described method [33]. The grains were rinsed with distilled water, and then ground using a blender for 2.5 min. The suspensions were passed through a 100-mesh sieve. The materials left on the screen were homogenized for 1.5 min again, and then passed through the same sieve. The starch–protein slurry was collected in a 1000-mL wide-neck flask and allowed to stand for 4 h. The supernatant was removed through suction and the settled starch layer was collected in 50 mL centrifuge tubes and centrifuged at 3000× *g* for 10 min. The upper non-white layer was scooped. The white layer was resuspended in distilled water and stirred for 30 min before centrifugation. The isolation procedures were repeated three times. The starch was then collected and dried in an oven at 40 °C for 48 h. Protein, lipid and ash contents in the isolated starch were lower than 4, 2, and 2 mg/g, respectively, indicating that starch purity reached the Chinese National Standard of edible maize starch (GB/T 8885−2017).

### 2.3. Starch Granule Size Distribution

The distribution of SGS was analyzed using a laser diffraction particle size analyzer (Mastersizer 2000, Malvern, Worcestershire, England). Instrument accuracy was verified with Malvern standard glass particles. The disperse phase was absolute ethyl alcohol. Size distribution was expressed in terms of the volume of equivalent spheres, and the granule size was defined as the volume weighted mean [34]. 

### 2.4. Amylopectin Chain Length Distribution

Starch (5 mg) was dissolved in 5 mL distilled deionized water in a boiling water bath for 60 min. Sodium azide solution (10 µL 2%*w*/*v*), acetate buffer (50 µL, 0.6 M, pH 4.4), and isoamylase (10 µL, 1400 U, EC 3.2.1.68, Sigma, St. Louis, MI, USA) were added to the starch dispersion, and the mixture was incubated in a water bath at 37 °C for 24 h. The hydroxyl groups of the debranched glucans were reduced by treatment with 0.5% (*w*/*v*) of sodium borohydride under alkaline conditions for 20 h. The preparation of about 600 µL was dried in vacuo at room temperature and allowed to dissolve in 20 µL of 1 m NaOH for 60 min. Then, the solution was diluted with 580 µL of distilled water. 

The chain length distribution of debranched amylopectin was determined using an ion chromatograph (ICS5000, Thermo Fisher Scientific, Waltham, MA, USA) in accordance with a previously published protocol [34,35]. 

### 2.5. Molecular Weight Distribution

The starch was debranched with isoamylase (EC 3.2.1.68, Sigma) dissolved in 50 mM sodium acetate, and its molecular weight distribution was analyzed following a method described previously using a PL-GPC 220 high-temperature chromatograph (Agilent Technologies UK Limited, Shropshire, UK) with three columns (PL110-6100, 6300, and 6525) and a differential refractive index detector [36,37]. The eluent system was dimethyl sulfoxide (DMSO) containing 0.5 mM NaNO_3_ at a flow rate of 0.8 mL/min. The column oven temperature was controlled at 80 °C. Standard dextrans of known molecular weights (2800, 18,500, 111,900, 410,000, 1,050,000, 2,900,000, and 6,300,000) were used for column calibration.

### 2.6. X-ray Diffraction Pattern

The X-ray diffraction patterns of starch were obtained using an X-ray diffractometer (D8 Advance, Bruker-AXS, Karlsruhe, Germany) operated at 200 mA and 40 kV. The scanning region of the diffraction angle (2*θ*) ranged from 3° to 40° at a step size of 0.04° with a count time of 0.6 s. Before measurements, the specimens were stored in a moist chamber where a saturated solution of NaCl maintained a constant humidity (relative humidity = 75%) for 1 week. RC (%) was calculated as the percentage ratio of the sum of the total crystalline peak areas to that of total diffractogram (sum of total crystalline and amorphous peak areas) using MDI Jade 6 software.

### 2.7. Pasting Properties

The pasting properties of starch (28 g total weight; 7%, *w*/*w*, dry weight) were evaluated using a rapid viscosity analyzer (RVA, Model 3D, Newport Scientific, Warriewood NSW, Australia) in accordance with a previously described method [34]. A sample suspension was equilibrated at 50 °C for 1 min, heated to 95 °C at 12 °C/min, maintained at 95 °C for 2.5 min, cooled to 50 °C at 12 °C/min, and maintained at 50°C for 1 min. The paddle speed was set at 960 rpm for the first 10 s and then decreased to 160 rpm for the rest of the analysis.

### 2.8. Thermal Properties

The thermal characteristics of the starch were determined using a differential scanning calorimeter (DSC, Model200*F3* Maia, NETZSCH, Bavaria, Germany) in accordance with a previously described method [34]. Each sample (5 mg, dry weight) was loaded into an aluminum pan (25/40 mL, d = 5 mm), and 10 mL of distilled water was added to achieve a starch–water suspension containing 66.7% water. The samples were hermetically sealed and allowed to stand at 4 °C for 24 h before they were heated in the DSC analyzer. The DSC analyzer was calibrated using an empty aluminum pan as a reference. The pans were heated at a rate of 10 °C/min from 20 to 100 °C. The thermal transitions of starch were defined as onset temperature (*T*_o_), peak gelatinization temperature (*T*_p_), conclusion temperature (*T*_c_), and gelatinization enthalpy (Δ*H*_gel_). After the thermal analysis was conducted, the samples were stored at 4 °C for 7 days for retrogradation investigations. Retrogradation enthalpy (Δ*H*_ret_) was automatically evaluated, and retrogradation percentage (*%R*) was calculated as *%R* = 100 × Δ*H*_ret_/Δ*H*_gel_.

### 2.9. Statistical Analysis

Values shown in all tables and figures are means of three independent ears. Means were compared by analysis of variance (ANOVA) with the least significant difference test at the 0.05 probability level using Data Processing System 7.05 (DPS, version 7.05, Hangzhou, China).

## 3. Results and Discussion

### 3.1. Starch Granule Size

The size distribution of starch granules of all the samples presents dual peaks. In comparison to 0F, the SGS was enlarged by fertilization (CF and SF) in both hybrids (Figure 1). Meanwhile, the SGS of SF was similar to CF in both hybrids. Fertilization increased the SGS in Tartary buckwheat [11], whereas the proportion of large granules in common and Tartary buckwheat decreased by N fertilization [8,10]. Under the same fertilization level, the SGS gradually enlarged with the increase of the ratio of N fertilization at the panicle initiation stage [18], and postpone the N application time enlarged the SGS by increase the proportion of medium size granules [14]. The similar SGS between the SF and CF indicated that a one-off application of SF can achieve the same effect to the three applications of conventional fertilizer and can save the two-time fertilization labor cost. Studies on wheat [7] and rice [38] have also observed that the fertilization favored the formation of small starch granules. The discrepancy may be due to different starch granule development dynamics, the starch development in wheat first developed A-type starch granules followed by B-type starch granules [39], and the fertilization offered enough resources to develop B-type starch granules. In the maize endosperm, the starch develops with an increase in granule numbers in the first two weeks and enlarged granule sizes after that [40]. 

### 3.2. Molecular Weight Distribution

The isoamylase debranched starch presented trimodal peaks for all the samples (Figure 2), with peak 3 representing the amylose content, and peaks 1 and 2 representing the percentages of amylopectin short chains and long chains. The ratio of peak 1/peak 2 is commonly regarded as the branching degree of amylopectin [17]. Studies on common and Tartary buckwheat have reported that the amylose content was decreased by N fertilization [8,10]. A study on rice observed that the peak 3 ratio in response to N rates at the late stage was dependent on the genotypes [17]. In the present study, the amylose content (peak 3 ratio) was unaffected by CF and decreased by SF in both hybrids (Table 1). The peak 2 ratio was increased by fertilization and the value was similar between CF and SF in both hybrids. The peak 1 ratio was unaffected by CF and increased by SF in JY877 and the value was decreased by CF and unaffected by SF in SY30. The peak 1/peak 2 ratio was reduced by CF and unaffected by SF in JY877, whereas the value was decreased by fertilization and the decrease was severe under CF in SY30, indicating that CF with large SGS has a high ratio of amylopectin with long chains [41]. This observation is consistent with a previous report on rice with late-stage N application [17].

### 3.3. Amylopectin Chain Length Distribution

The average ACL of isoamylase-debranched amylopectin was decreased by the fertilization and the influence of this was more significant in JY877 (Figure 3). For JY877, the ratios of degree of polymerization (DP) with 6–9 and 22–76 were decreased and DP 10–21 was increased by fertilization, with the influence being higher for CF than for SF. For SY30, the ACL was similarly decreased in both SF and CF by fertilization, but the difference ratio was lower than 0.17% for DP 6–76, indicated that it was less affected by fertilization than JY877. In Tartary buckwheat, the ACL was increased by fertilization [11]. In rice, with the increase of N rate, the ACL first decreased and then increased, resulting in the highest content in short chain distribution under a moderate N rate [15,16,42]. Another study has reported that ACL is unaffected by N fertilization [43], and that the ACL gradually increased with the increase of N rate at the panicle stage [44]. In waxy maize, the ACL was unaffected by CF but its response to SF was different between the two years [30]. A study on rice has reported that the ACL’s response to N rates is dependent on cultivars, and that the change is consistent to the relative expression level of soluble starch synthase I and branching enzymes IIb [12].

### 3.4. XRD Pattern

The XRD pattern of both hybrids under all the treatments presents a typical “A”-type pattern (Figure 4). The RC in JY877 was increased by fertilization and the increase was higher for CF than for SF, whereas the value in SY30 was unaffected by SF and increased by CF. The high RC under CF may be due to how the starch has a large SGS and a low peak 1/peak 2 ratio [41]. A study on wheat [7], Tartary buckwheat [9], common buckwheat [8], and rice [18,38] also observed that the fertilization increases the RC, whereas a study on ‘super’ rice reported that the RC was dependent on N rates [15]. Meanwhile, a study has observed that RC in response to N fertilization was different among various cultivars [13]. Under the same fertilization level, the RC gradually increased with the increase of the ratio of N fertilization at the panicle initiation stage [18], and postponing the N application time increased the RC [14].

### 3.5. Pasting Property

Pasting temperature (*P*_temp_) and SB in SY30 were unaffected by fertilization, those two values in JY877 were unaffected by CF, and the *P*_temp_ was decreased whereas the SB was increased by SF (Table 2). For JY877, the peak viscosity (PV) and TV were increased by fertilization similarly between SF and CF, the FV was increased by fertilization and the increase was higher for SF than for CF, and breakdown viscosity (BD) was unaffected by CF and decreased by SF. For SY30, PV, TV, BD, and FV were increased by fertilization and the increases were higher for CF than for SF. A study on rice has reported that the starch PV, TV, and BD were higher under SF [23]. The pasting viscosities were decreased by the N fertilization in Tartary buckwheat [11], common buckwheat [8,9], triticale [45], and rice [12,13,38]. In ‘super’ rice, the pasting viscosities were increased by low N rate and decreased by excessive N rate [15]. Under the same fertilization level, PV, TV, BD, and FV gradually increased with an increase of the ratio of N fertilization at the panicle initiation stage [12], but those parameters decreased with the postponement of the N application time [14]. Between the two hybrids, SY30 has higher PV, TV, and BD than that of JY877 with fertilization. 

### 3.6. Thermal Property

The Δ*H*_gel_ in JY877 was unaffected by CF and reduced by SF, whereas in SY30 it was decreased by fertilization in a similar way for both CF and SF. For JY877, *T*_o_ and *T*_p_ were increased by fertilization and the increases were similar between both CF and SF (Table 3). *T*_c_ was unaffected by SF and reduced by CF. For SY30, *T*_p_ and *T*_c_ were unaffected and decreased by fertilization, *T*_o_ was increased by CF and decreased by SF. In Tartary buckwheat starch, the gelatinization temperatures were decreased by low N rate but increased by moderate and high N rates, Δ*H*_gel_ was increased by N application [9,10]; whereas Δ*H*_gel_ and gelatinization temperatures of common buckwheat were increased by N fertilization [8]. In rice, the gelatinization temperatures were unaffected by N rates, but Δ*H*_gel_ was discrepant between the cultivars [12]. Another study observed that Δ*H*_gel_ and gelatinization temperatures of cultivars with normal and low amylose contents were unaffected by N fertilization [13]. Under the same fertilization level, the Δ*H*_gel_ was increased and gelatinization temperatures were decreased under the higher ratio of N fertilization at the panicle initiation stage [18]. With a postponed application time of the N, the Δ*H*_gel_ was increased and gelatinization temperatures were decreased [14]. In ‘super’ rice, the Δ*H*_gel_ and gelatinization temperatures were decreased by a low N rate and increased by an excessive N rate [15]. 

In the present study, the Δ*H*_ret_ in SY30 was increased by fertilization, whereas in JY877 it was increased by CF and decreased by SF. The *%R* was increased by CF in both hybrids, whereas with SF it was decreased in JY877 and increased in SY30. The higher Δ*H*_ret_ and *%R* under CF may be due to how the starch has a larger SGS, higher RC, and lower peak 1/peak 2 ratio and ACL. A study on rice has observed that Δ*H*_ret_ and *%R* were lower in SF than in CF under the same fertilization level [23]. In ‘super’ rice, the Δ*H*_ret_ and *%R* were decreased by a low N rate and increased by an excessive N rate [15]. Another study has reported that the Δ*H*_ret_ and *%R* of rice starch was gradually decreased with increased N rates [38]. A study on rice also observed that the Δ*H*_ret_ and *%R* of cultivars with normal or low amylose contents were increased by N fertilization [13], but under the same fertilization level achieved a high ratio of N fertilization at the panicle initiation stage [18] while late N application time [14] induced lower Δ*H*_ret_ and *%R*. 

## 4. Conclusions

The starch physicochemical properties of normal maize were affected by fertilization modes. In comparison to 0F, larger SGS, higher RC, lower peak 1/peak 2 ratio and ACL were obtained under CF, which results in higher PV, TV, BD, *T*_o_, *T*_p_, Δ*H*_ret_ and *%R* in both hybrids. Larger SGS, lower amylose content and ACL, and higher PV, TV, and FV were obtained under SF in both hybrids, whereas the responses of other parameters were different between the two hybrids. In comparison to CF, the similar SGS, lower amylose content and RC, and higher peak 1/peak 2 ratio were obtained under SF in both hybrids, but ACL, and pasting and thermal properties were dependent on the hybrids. Between the two hybrids, JY877 has higher Δ*H*_gel_, lower pasting viscosities and *%R* than those of SY30 with fertilization, which may be due to how JY877 has lower amylose content, SGS and ACL and higher RC. The results offer the option to choose the optimal maize hybrid and fertilization mode based on different food utilizations. 

## Figures and Tables

**Figure 1 polymers-15-00083-f001:**
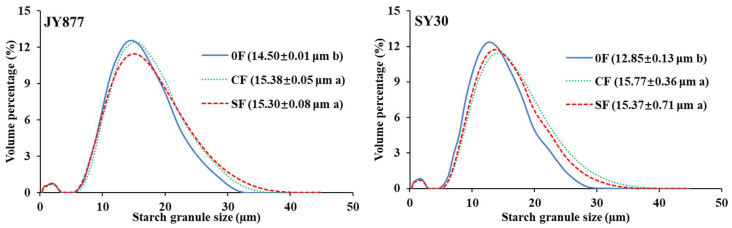
Effects of the fertilization mode on the starch granule size distribution of normal maize. The data in brackets are the average granule size. Mean values within the same hybrid followed by different letters are significantly different at *p* < 0.05. JY877, Jiangyu877; SY30, Suyu30; 0F, zero fertilization; CF, common fertilization; SF, slow-release fertilization.

**Figure 2 polymers-15-00083-f002:**
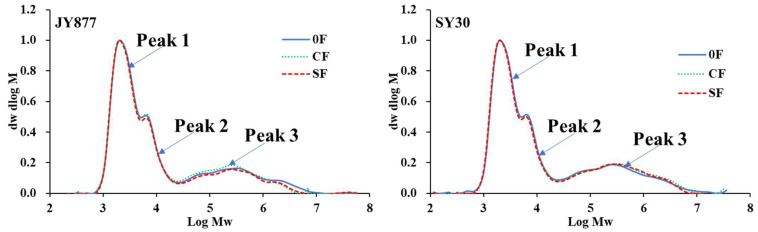
Effects of the fertilization mode on isoamylase-debranched starch GPC profiles of normal maize. JY877, Jiangyu877; SY30, Suyu30; 0F, zero fertilization; CF, common fertilization; SF, slow-release fertilization.

**Figure 3 polymers-15-00083-f003:**
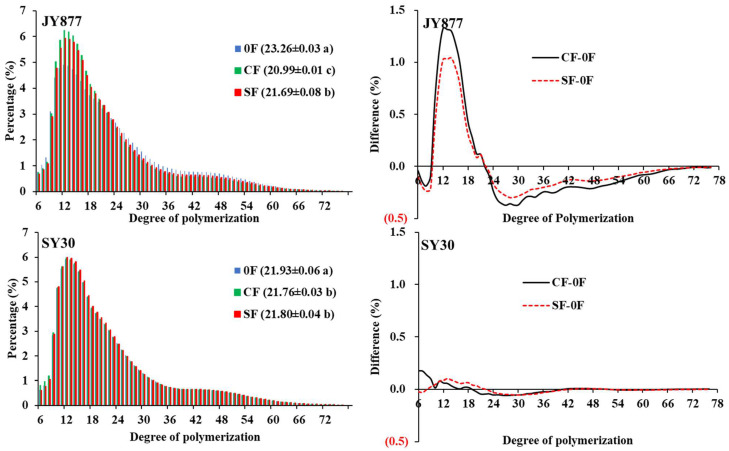
Effects of the fertilization mode on isoamylase-debranched amylopectin chain length distribution of normal maize. The data in brackets are the average chain length. Mean values within the same hybrid followed by different letters are significantly different at *p* < 0.05. JY877, Jiangyu877; SY30, Suyu30; 0F, zero fertilization; CF, common fertilization; SF, slow-release fertilization.

**Figure 4 polymers-15-00083-f004:**
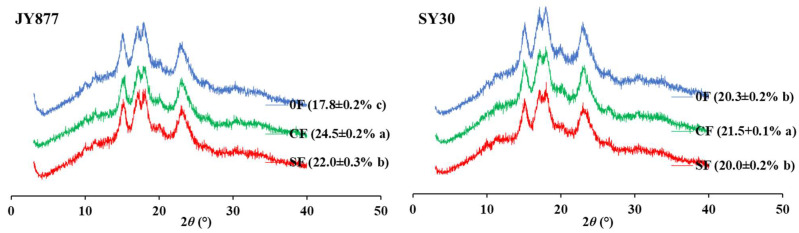
Effects of the fertilization mode on starch X-ray diffraction pattern of normal maize. The Data in brackets are the relative crystallinity. Mean values within the same hybrid followed by different letters are significantly different at *p* < 0.05. JY877, Jiangyu877; SY30, Suyu30; 0F, zero fertilization; CF, common fertilization; SF, slow-release fertilization.

**Table 1 polymers-15-00083-t001:** Effects of the fertilization mode on percentage of three peaks of isoamylase-debranched starch of normal maize.

Hybrid	Fertilization	Peak 1 (%)	Peak 2 (%)	Peak 3 (%)	Peak 1/Peak 2
JY877	0F	53.12 ± 0.22 b	19.58 ± 0.43 b	27.30 ± 0.21 c	2.71 ± 0.07 b
	CF	52.91 ± 0.06 b	20.22 ± 0.24 a	26.87 ± 0.18 c	2.62 ± 0.03 c
	SF	54.48 ± 0.00 a	20.13 ± 0.21 a	25.39 ± 0.21 d	2.71 ± 0.03 b
SY30	0F	50.48 ± 0.01 c	18.26 ± 0.79 d	31.25 ± 0.79 a	2.77 ± 0.12 a
	CF	49.69 ± 0.10 d	18.92 ± 0.09 c	31.39 ± 0.19 a	2.63 ± 0.01 c
	SF	50.81 ± 0.21 c	18.79 ± 0.07 c	30.40 ± 0.14 b	2.70 ± 0.02 b

Mean values in the same column followed by different letters are significantly different at *p* < 0.05. JY877, Jiangyu877; SY30, Suyu30; 0F, zero fertilization; CF, common fertilization; SF, slow-release fertilization.

**Table 2 polymers-15-00083-t002:** Effects of the fertilization mode on starch pasting properties of normal maize.

Hybrid	Fertilization	PV (mPa.s)	TV (mPa.s)	BD (mPa.s)	FV (mPa.s)	SB (mPa.s)	*P*_temp_ (°C)
JY877	0F	672 ± 31 d	622 ± 28 e	50 ± 3 b	668 ± 27 e	46 ± 4 c	86.5 ± 1.3 a
	CF	735 ± 12 c	687 ± 11 c	47 ± 0 b	735 ± 13 d	48 ± 1 c	87.0 ± 0.4 a
	SF	728 ± 14 c	698 ± 16 c	30 ± 2 c	869 ± 17 a	170 ± 11 a	79.1 ± 0.4 b
SY30	0F	689 ± 14 d	655 ± 11 d	34 ± 3 c	786 ± 10 c	130 ± 5 b	78.0 ± 0.0 b
	CF	864 ± 5 a	792 ± 10 a	72 ± 8 a	910 ± 14 a	117 ± 24 b	79.3 ± 1.0 b
	SF	769 ± 67 b	716 ± 62 b	54 ± 9 b	818 ± 68 b	103 ± 14 b	79.1 ± 1.0 b

Mean values in the same column followed by different letters are significantly different at *p* < 0.05. JY877, Jiangyu877; SY30, Suyu30; 0F, zero fertilization; CF, common fertilization; SF, slow-release fertilization. PV, peak viscosity; TV, trough viscosity; BD, breakdown viscosity; FV, final viscosity; SB, setback viscosity; *P*_temp_, pasting temperature.

**Table 3 polymers-15-00083-t003:** Effects of the fertilization mode on starch thermal properties of normal maize.

Hybrid	Fertilization	Δ*H*_gel_ (J/g)	*T*_o_ (°C)	*T*_p_ (°C)	*T*_c_ (°C)	Δ*H*_ret_ (J/g)	*%R* (%)
JY877	0F	10.3 ± 0.3 b	68.3 ± 0.2 c	73.4 ± 0.1 b	80.9 ± 0.2 a	4.3 ± 0.2 b	41.9 ± 1.3 c
	CF	10.3 ± 0.4 b	69.4 ± 0.1 a	74.0 ± 0.0 a	79.0 ± 0.4 b	5.4 ± 0.1 a	52.4 ± 3.3 b
	SF	9.6 ± 0.5 c	69.6 ± 0.2 a	74.4 ± 0.0 a	80.2 ± 0.4 a	3.2 ± 0.2 d	32.8 ± 1.1 e
SY30	0F	11.4 ± 0.5 a	69.1 ± 0.2 b	73.9 ± 0.1 ab	80.8 ± 0.5 a	3.8 ± 0.2 c	35.8 ± 0.3 d
	CF	8.7 ± 0.1 d	69.5 ± 0.0 a	74.2 ± 0.0 a	78.8 ± 0.1 b	5.4 ± 0.1 a	61.2 ± 1.8 a
	SF	8.8 ± 0.8 d	68.5 ± 0.6 c	73.6 ± 0.3 b	78.7 ± 0.2 b	5.2 ± 0.3 a	59.4 ± 2.5 a

Mean values in the same column followed by different letters are significantly different at *p* < 0.05. JY877, Jiangyu877; SY30, Suyu30; 0F, zero fertilization; CF, common fertilization; SF, slow-release fertilization. Δ*H*_gel_, gelatinization enthalpy; *T*_o_, onset temperature; *T*_p_, peak gelatinization temperature; *T*_c_, conclusion temperature; Δ*H*_ret_, retrogradation enthalpy; *%R*, retrogradation percentage.

## Data Availability

All the data and code used in this study can be requested by email to the corresponding author Dalei Lu at dllu@yzu.edu.cn.

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
