# Peer review of "Starch Physicochemical Properties of Normal Maize under Different Fertilization Modes"

_polymers, 2022, doi:10.3390/polym15010083_

Round 1

Reviewer 1 Report

the article is not a research article, it is an article presenting results.  It is suggested to the author to rewrite the article giving the research approach. Which variable is the most important (input variable) that modifies the response variables and propose the model or mechanism to determine how it modifies them.?

Author Response

 Following the suggests, the introduction was improved (the first paragraph introduced the N application on grain quality, the second paragraph introduced the slow-release fertilizer on grain yield and quality, and the third paragraph point out the significance of the MS); the references were checked carefully (a repeated reference was deleted and a new published relative paper was added); the research design and method was checked and improved; and the results and discussion was improved as well as we can. In the conclusion section, we proposed the relation between the structure and physicochemical characteristics, and the difference between SF and CF, which can offer the reference in maize production.

Reviewer 2 Report

The manuscript is presented briefly and clearly. In the Introduction, the facts that provide a preface to the research and explain the reason for conducting the study are briefly and clearly given. This research has a scientific basis with a clearly presented experimental design in the Materials and Methods. In my opinion, this study was designed and carried out to appropriate technical standards.

My correction suggestions are:

Line 185, 206, 225, 240, 261: The hybrids (JY877 and SY30) are clearly indicated on the Graphs and Tables, so, in my opinion, there is no need to subsequently emphasize which hybrids were studied. But it is not a problem to keep it like that, just in that case correct SY30 in Suyu30 because SY30 is repeated twice. If you decide on this option, please add below Table 3.

Line 189-190: Figure 1, probably not Figure S1

Line 251: (Gao et al., 2021) – please put the reference number, as well as the others

Line 252: t(3.4) – please delete t

Line 173, 214-215, 231: You present the results obtained on tartar buckwheat and as a reference, you cite No. 13, which refers to japonica rice. Please check the references.

Author Response

Line 185, 206, 225, 240, 261: The hybrids (JY877 and SY30) are clearly indicated on the Graphs and Tables, so, in my opinion, there is no need to subsequently emphasize which hybrids were studied. But it is not a problem to keep it like that, just in that case correct SY30 in Suyu30 because SY30 is repeated twice. If you decide on this option, please add below Table 3.

Response: Revised as suggested.

Line 189-190: Figure 1, probably not Figure S1.

Response: The Figure was inset as figure 2 in the MS.

Line 251: (Gao et al., 2021) – please put the reference number, as well as the others

Response: Revised as suggested.

Line 252: t(3.4) – please delete t

Response: Revised as suggested.

Line 173, 214-215, 231: You present the results obtained on tartar buckwheat and as a reference, you cite No. 13, which refers to japonica rice. Please check the references.

Response: We checked the MS and the reference in Lines 214-215, and 231 was replaced by No. 7, and the reference in line 173 should be No. 13. And all the references were checked throughout the MS.

Reviewer 3 Report

Comments and Suggestions for Authors

The manuscript entitled " Effects of fertilization mode on the starch physicochemical properties of normal maize" is based on original research experiment and the presented results therein broaden the knowledge in the field of applied plant science and agronomy. The publication presents some interesting studies. The paper is generally good organized, presented in a logical sequence, and has adequate bibliographic review. This work can be a source of some information.

However, the text needs to be thoroughly rewritten before the publication of this work:

1) Key words: they must be different from those used in the title of the work.

2) The language should be carefully checked by a native speaker.

3) The introduction can be improved and done with more care for this topic

Author Response

The manuscript entitled " Effects of fertilization mode on the starch physicochemical properties of normal maize" is based on original research experiment and the presented results therein broaden the knowledge in the field of applied plant science and agronomy. The publication presents some interesting studies. The paper is generally good organized, presented in a logical sequence, and has adequate bibliographic review. This work can be a source of some information.

However, the text needs to be thoroughly rewritten before the publication of this work:

 Key words: they must be different from those used in the title of the work.

Response: the key words “normal maize”, “fertilization mode” and “thermal property” were deleted and “slow-release fertilizer” and “retrogradation” were added. 

2) The language should be carefully checked by a native speaker.

Response: the language was carefully checked and improved as well as we can.

3) The introduction can be improved and done with more care for this topic

 Response: The introduction was improved with more focused on the topic. The first paragraph introduced the N application on grain quality, the second paragraph introduced the slow-release fertilizer on grain yield and quality, and the third paragraph point out the significance of the MS.

Round 2

Reviewer 1 Report

According with the abstract, the conclusion is that the fertilization mode change physicochemical properties of two normal maize. This conclusion not fit with the focus of polymers journals. Isnt it?

Discussion and interpretation of results are based in the fertilization mode and how fertilization mode change normal maize properties. These results are important for agronomic researchers. Depending of fertilization mode is possible to have different size of normal maize.

The author should change the focus, focusing on starches and not on fertilization modes. For example, the author shows that he obtained a JY877 starch that has higher viscosity than a control. So, I write an article to Polymers where I report a modified starch that has a higher viscosity and I show its physicochemical properties and my job is to find out why it has more viscosity. And the answer is because the mode of fertilization decreased the average amylopectin chain length allowing a higher viscosity.

Author Response

Comment 1: According with the abstract, the conclusion is that the fertilization mode change physicochemical properties of two normal maize. This conclusion not fit with the focus of polymers journals. Isnt it?

Response: In the abstract, the significance of the study was addressed in the first sentence, and the conclusion was revised based on the starch was desired function under different fertilization mode. We hope this revision can meet the focus of the Journal. 

Comment 2: Discussion and interpretation of results are based in the fertilization mode and how fertilization mode change normal maize properties. These results are important for agronomic researchers. Depending of fertilization mode is possible to have different size of normal maize.

Response: In the introduction section, the importance of starch with desired functions was addressed. Some relevant references were added, and some depictions about the N rate on starch quality of other cereal crops and SF fertilization on grain quality were simplified.

In M&M section, some details about the analysis method were depicted in detail.

In Results and Discussion section, some discussion was added.

In conclusion section, the depiction was focused on the starch.

Comment 3: The author should change the focus, focusing on starches and not on fertilization modes. For example, the author shows that he obtained a JY877 starch that has higher viscosity than a control. So, I write an article to Polymers where I report a modified starch that has a higher viscosity and I show its physicochemical properties and my job is to find out why it has more viscosity. And the answer is because the mode of fertilization decreased the average amylopectin chain length allowing a higher viscosity.

Response: Following the advice, we change the focus from fertilization mode to starch with desired functions. For example, in the conclusion section, we depicted the starch with desired functional character may be obtained under SF, which can result in higher PV, TV, BD, To, Tp, ΔHret and %R. The results offer the option to choose the optimal maize hybrid and fertilization mode based on different food utilizations.

Round 3

Reviewer 1 Report

Incorporate the standard deviation in figure 1. Do statistical analysis to determine if the granule size is significantly different

According with figure 1, Between all volumes it gives 45% , then, where is the rest of  volume or the rest of particle size.  What does means “ volume percentage” in figure 1? There are only 12% de granules of 14.5 um?  The figure and the information is not understandable

Line 163. Dual peaks? Where is the dual peak in the fig 1.? I only see one peak. Discuss in text.

Line 165. Calculate standard deviation to affirm that is smaller. Discuss in text.

Line 65. There is a difference of 0.08 um between CF and SF in JY877 and  0.4 um in SY30. Why author write: “Smaller”? Discuss in text.

Figure 2. I do not see the difference. I see the same value of  y axis for 0F, CF y SF.

Which are the units for y axis?

Table 1. It is not possible that author shows letters and not show the standard deviation. Add standard deviation. Discuss in text.

Units for figure 2 and units in table 1 not correspond. Homogenize units for clarity in the results.

 Figure 3. I do not see difference between treatments. Why author make a subtraction between treatments?

Figure 4.  In figure the diffraction peaks for CF and SF have less amplitude. I think that author do not use the same criteria to calculate RC.  It is very difficult to explain a difference of 1% or 2% in DRX measurements.

Table 2. Author must add deviation standard.

Table 3. Author must add deviation standard.
